# Power generator driven by Maxwell's demon

Kensaku Chida[1], Samarth Desai[1], Katsuhiko Nishiguchi[1] & Akira Fujiwara[1]

Maxwell's demon is an imaginary entity that reduces the entropy of a system and generates free energy in the system. About 150 years after its proposal, theoretical studies explained the physical validity of Maxwell's demon in the context of information thermodynamics, and there have been successful experimental demonstrations of energy generation by the demon. The demon's next task is to convert the generated free energy to work that acts on the surroundings. Here, we demonstrate that Maxwell's demon can generate and output electric current and power with individual randomly moving electrons in small transistors. Real-time monitoring of electron motion shows that two transistors functioning as gates that control an electron's trajectory so that an electron moves directionally. A numerical calculation reveals that power generation is increased by miniaturizing the room in which the electrons are partitioned. These results suggest that evolving transistor-miniaturization technology can increase the demon's power output.

[1] NTT Basic Research Laboratories, NTT Corporation, 3-1 Morinosato Wakamiya, Atsugi, Kanagawa 243-0198, Japan. Correspondence and requests for materials should be addressed to K.C. (email: chida.kensaku@lab.ntt.co.jp).

One famous example of energy generation by Maxwell's demon is when it observes individual randomly moving gas particles in a box[1-3]. As feedback responding to the observation, the demon opens and closes a gate separating the box into two areas so that colder and hotter gas particles are sorted out. Thanks to such observation and the following feedback, the temperature in one area differs from that in the other, which means free energy is generated and entropy is reduced in the box. The noteworthy feature of this energy generation is that, without friction, Maxwell's demon does not consume energy during the feedback. This gives rise to a question regarding a violation of the second law of thermodynamics because the demon looks like it is producing energy from a thermal bath without any dissipation.

This question has been answered by the concept of information thermodynamics[4-13]: The generated free energy originates from the demon's memory. The demon uses energy during a cycle of the memory operation for storing the information about randomly moving objects at the measurement and erasing it after the feedback[4,5]. Consequently, the energy it generates perfectly follows the second law of thermodynamics: When Maxwell's demon generates energy, a corresponding dissipation takes place in the demon's memory.

In addition to this theoretical progress, technical advances allowing access to a small system in which a few small objects are thermally fluctuating has led to experimental demonstrations of Maxwell's demon[14-23], by using rotaxane molecules[14] and beads a few-hundred nanometers in diameter[15]. Using electrons, the demon has also realized a Szilard's engine that generates free energy[18]. Further, an autonomous Maxwell's demon that enables automatic observation of and feedback to electrons has been recently demonstrated using a chip integrated with superconductor devices, and the thermodynamics of the demon itself was revealed[20]. In addition, some previous experiments have succeeded in extracting work[15,20] and power[22] with the demon. Here we present a demonstration of one of the most natural possible Maxwell's demon that outputs electric power by rectifying randomly moving electrons in transistors.

## Results

**Device structure and the demon's feedback protocol.** The power generation by the demon is carried out on a silicon transistor chip integrating a small box (hereafter referred to as a single-electron box (SEB)) and a charge sensor that monitors electron motion (Fig. 1a,b, also see Methods section and Supplementary Note 1). The SEB is separated from the source and drain by two transistors, G1 and G2, which function as gates for electrons. The number of electrons $n$ in the SEB is around 100 on average. It fluctuates due to thermal agitation (Fig. 1c), which is monitored in real time with the sensor used for the feedback, which is explained next.

When G1 opens and G2 closes, electrons shuttle randomly between the source and SEB due to thermal fluctuation (hereafter referred to as state A). When $n$ increases by $\Delta n_{\text{thresh}}$ larger than zero from $n$ observed initially at state A, G1 and G2 are closed and open, respectively, and then electrons shuttle randomly between the drain and SEB (state B). When $n$ decreases by $\Delta n_{\text{thresh}}$ from $n$ observed initially at state B, G1 and G2 are again open and closed, respectively (state A). This feedback cycle comprising state A and B transfers electrons from the source to drain (Fig. 1d). In other words, the feedback cycle rectifies the random motion of electrons and converts it into electric current $I_{\text{MD}}$ ( $= \langle \Delta n_t \rangle \times e/\Delta t_{\text{cyc}}$, where $\langle \Delta n_t \rangle$ is the ensemble average of the number of transferred electrons, $e$ is the elementary charge and $\Delta t_{\text{cyc}}$ is the time taken for one cycle in which the state changes from A to B and back to A). Here, $I_{\text{MD}}$ is defined as a current flowing from the drain to source (forward current). As can be easily expected, when $\Delta n_{\text{thresh}}$ is negative, current flows in the opposite direction (backward current). When source–drain bias voltage $V_{\text{SD}}$ ( $= V_{\text{S}} - V_{\text{D}}$) is applied across the source and drain, the electric power the demon generates can be evaluated to be $- I_{\text{MD}} \times V_{\text{SD}}$, where the minus sign represents the current direction against $V_{\text{SD}}$, or electrons climbing up the chemical potential.

Our silicon chip has two unique features that affect the demon's performance. The first feature is that the transport of electrons between the SEB and source or drain shows directionally asymmetric characteristics[24] because the transport is dominated by thermal hopping over the energy barrier induced by the transistor's gate (Fig. 2a). The transition rates for this thermal hopping is tunable by the gate voltage and proportional to $\exp(- \Delta E/k_{\text{B}}T)$, where $k_{\text{B}}$ is Boltzmann's constant, $T$ is temperature and $\Delta E$ is the height of the energy barrier the electrons have to surmount. For electrons in the source and drain, $\Delta E$ is the difference between the top of the energy barrier and chemical potential $\mu_{\text{S}}$ and $\mu_{\text{D}}$ of the source and drain, respectively. Therefore, the transition rate for the electrons to enter the SEB is constant because voltages applied to the transistors' gates, source and drain are constant during the measurement of electron motion. On the other hand, for electrons in the SEB, $\Delta E$ is the difference between the top of the energy barrier and the energy level of an electron in the SEB. This energy level is given by $\mu_{\text{SEB}} + (k - 0.5)e/C$, where $\mu_{\text{SEB}}$ and $C$ are the chemical potential and capacitance of the SEB, respectively, and $k$ is the deviation of $n$ from the integer part of the average of thermally fluctuating $n$. Since the energy level varies according to $k$, the rate at which an electron leaves the SEB varies with $k$ (Fig. 2b, the details are explained in Supplementary Note 3)[24]. Therefore, electrons move in a directionally asymmetric way, which is beneficial for power generation by the demon as explained later.

The second unique feature of our chip is that undesirable work applied to the system during the feedback process can be eliminated. One of the most important conditions for the feedback from the demon is that no work be applied to observed objects, but it is difficult to meet this condition in experiments because of the difficulty in partitioning the objects by a gate. Indeed, in previous reports[15,18-20], work was applied to the objects and then carefully considered as a net value to discuss information-originated free energy. In our chip, since transistors partition electrons using gates as mentioned above, the demon can, in principle, perform the feedback process without work exertion. However, in reality, since the transistors tuned by applied voltage $V_{\text{G1}}$ and $V_{\text{G2}}$ are capacitively coupled to the SEB, the change in $V_{\text{G1}}$ and $V_{\text{G2}}$ at the switch between state A and B shifts the SEB's chemical potential $\mu_{\text{SEB}}$, and thus an electron in the SEB can gain or lose energy. This energy shift, $\Delta\mu_{\text{SEB}}$, generates current even without the feedback by the mechanism of a single-electron ratchet[25] (the details are explained in Supplementary Note 4). To eliminate $\Delta\mu_{\text{SEB}}$, in our operation scheme, we cover the whole area with another gate (hereafter referred to as an upper gate (UG)). The UG is capacitively coupled to the SEB, which enables us to control $\mu_{\text{SEB}}$ at state A and B separately. When state A and B are alternately switched periodically without the feedback[26], even when voltage between the source and drain is zero, current flows due to the ratchet mechanism. This current can be tuned by tuning the UG-voltage difference $\Delta V_{\text{UG}}$ between state A and B, that is, $\Delta V_{\text{UG}} = V_{\text{UG,A}} - V_{\text{UG,B}}$, where $V_{\text{UG,A(B)}}$ is UG voltage at the state A(B) (Fig. 2c). With the difference $\Delta V_{\text{UG}} = 0.4$ V, no current is generated with the change between state A and B without feedback, which means that the UG enables us to eliminate

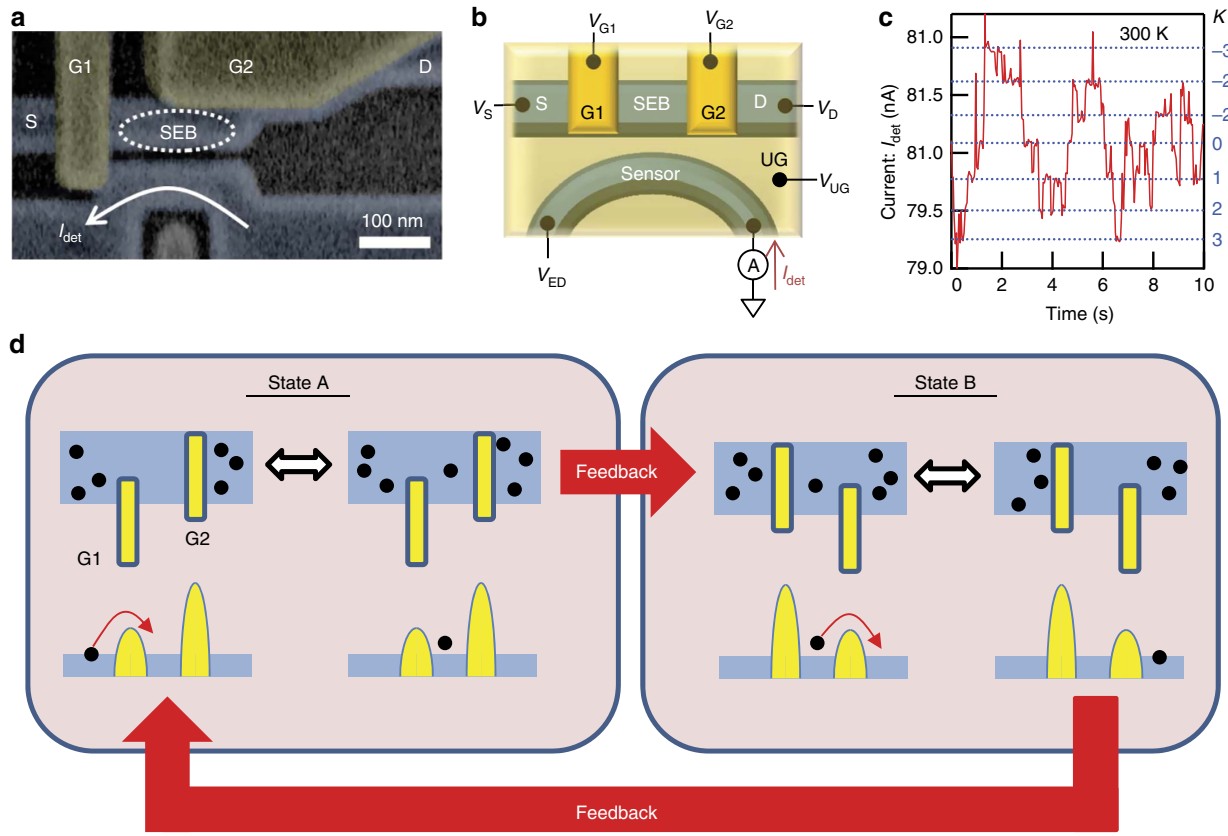

**Figure 1 | Device structure and operation schematics for Maxwell's demon.** (**a**) False-color scanning electron microscope image and (**b**) schematics of the fabricated device. Above a silicon wire channel covered with a silicon dioxide, two gate terminals are formed to make field-effect transistors Gate(G) G1 and G2. The single-electron box (SEB) is electrically formed between the source (S) and drain (D) because G1 and G2 form energy barriers between the S and D as shown in **d**. Another channel functioning as a charge sensor is located close to the SEB. Above the whole area, there is an upper gate (UG), to which positive voltage $V_{UG}$ is applied. The details are explained in Methods section. (**c**) Single-electron detection at room temperature. Current flowing through the sensor is monitored and the abrupt reduction (increase) of current with the same step height means that a single-electron enters (leaves) the SEB. $k$ is the deviation from the average of the number of electrons $n$ in the SEB. We applied $V_{UG} = 3.05$ V, $V_{ED} = 1$ V, $V_{G1} = -2.4$ V, $V_{G2} = -1.95$ V and $V_S = V_D = -0.6$ V. (**d**) Schematics of current generation by rectifying randomly moving electrons. The four upper and lower illustrations are schematics and energy-band diagrams along the S, SEB and D, respectively. For simplicity, they show how one electron is transferred, meaning $\Delta n$ explained in the main text is one. We define state A(B) as the case that G1 opens (closes) and G2 closes (opens). When G1 opens (closes), its energy barrier lowers (rises) and an electron shuttles between the S and SEB faster (slower).

undesirable work in the demon's feedback and monitor pure current generated by the demon.

**Power generation by Maxwell's demon.** All measurements were carried out at room temperature. The time interval $\Delta t_m$ of each measurement was set to 60 ms, which is short enough to monitor electron motion or $k$. More importantly, from the viewpoint of the demon's performance, $\Delta t_m$ was a bit shorter than (or comparable to) the average time intervals for $n$ to increase by $\Delta n_{thresh}$ at state A and decrease by $\Delta n_{thresh}$ at state B, which leads to high power generation and relatively high efficiency for information-to-energy conversion as discussed later. Note that $k$ and $\Delta n_{thresh}$ are different: $k$ is the deviation in $n$ from the integer part of the average of $n$ ($k = n - n_{ave}$, where $n_{ave}$ is the integer part of the average of $n$) and $\Delta n_{thresh}$ is the deviation in $n$ from the initially observed $n$ in the state and the threshold value to perform feedback. In our experiment, $\Delta n_{thresh}$ is determined by threshold value of $I_{det}$. We set the threshold as $1.5\delta I_{det}$, where $\delta I_{det}$ is the change in $I_{det}$ induced by the motion of an electron. In this work, since $I_{det}$ showed discrete steps (Fig. 1c), the change in $I_{det}$ became larger than $1.5\delta I_{det}$ when $\Delta n_{thresh} = 2$.

With the feedback, even when $V_{SD} = 0$ V, $I_{MD}$ of about 20 zA flows from the drain to source (forward current) (Fig. 3). On the

other hand, no current is generated without the feedback (Fig. 2c). Current in the opposite direction (backward current) can also be generated when $\Delta n_{thresh} = -2$. In addition, even when negative (positive) $V_{SD}$ is applied to the drain, forward (backward) current counter flows from the drain (source) to source (drain). These results indicate that the demon generates current by rectifying randomly moving electrons. Also noteworthy is that Monte-Carlo simulation with consideration of the directionally asymmetric electron shuttling mentioned above (Supplementary Note 2) can reproduce the experimental results well, which means that the directionally asymmetric electron shuttling actually takes place in our silicon chip.

Generated power is given by $-I_{MD} \times V_{SD}$. When $V_{SD}$ is changed from 0 V in a negative direction, the generated power increases because of the increase in $V_{SD}$ and then decreases because of the reduction in $I_{MD}$ (Fig. 4a). This behaviour can also be reproduced by Monte-Carlo simulations. In our experiments, maximum power of about 0.5 zW was generated. Such small power and current can be evaluated thanks to real-time electron counting by the high-sensitivity charge sensor, which leads to this proof-of-concept demonstration of Maxwell's demon. On the other hand, the generated power can be increased, in principle, by reducing $\Delta t_m$ and the gate-tunable transition time for electrons to

hop over the energy barrier. However, from the technical perspective, $\Delta t_m$ is limited by the time resolution of the charge sensor. High-speed charge sensing, for example, using a

reflectometry method based on radio-frequency signals[27] or a faster feedback process using an on-chip circuit, can reduce $\Delta t_m$ and thus increase $I_{MD}$ and the power.

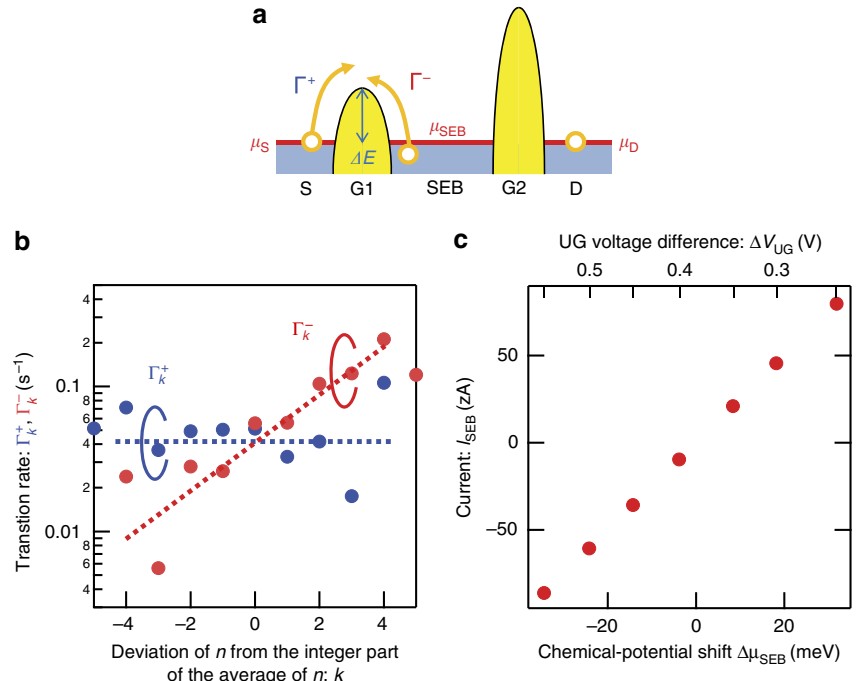

**Figure 2 | Directionally asymmetric electron transport in transistors. (a)** Energy-band diagram along the S and D, where $\mu_S$, $\mu_D$ and $\mu_{SEB}$ are chemical potentials at the source (S), drain (D) and single-electron box (SEB). For simplicity, we consider the electron shuttling between the S and SEB. $\Gamma^+$ and $\Gamma^-$ are transition rates from the S and SEB, respectively. $\Delta E$ is energy barrier height when an electron in the S surmounts the energy barrier. **(b)** Transition rates $\Gamma_k^+$ and $\Gamma_k^-$ as a function of $k$, which is the deviation in $n$ from the integer part of the average. They are the transition rates for electrons to enter ($+$) and leave ($-$) the SEB at $k$. The dotted lines show expected values. The details are explained in Supplementary Note 3. Since the number of data points at $|k| > 4$ is relatively small, there are large errors at $|k| \geq 4$. **(c)** Current across the SEB $I_{SEB}$ generated by periodic switching without feedback, estimated with single-electron counting statistics, as a function of the shift in $\mu_{SEB}$ ($\Delta\mu_{SEB}$) when the state is switched from A to B. The upper axis depicts $\Delta V_{UG}$, which is the difference in voltage applied to the upper gate (UG) when the state is switched from A to B. State A and B are alternately switched every 4 s. Current is given by $\Delta n e / 8$, where $\Delta n$ is the number of electrons transferred from S to D when the state is switched from A to B. The factor of 8 originates from the time it takes for the state changes from A to B and back to A, that is, $2 \times 4$ s. The details, including the evaluation of $\Delta\mu_{SEB}$, are explained in Supplementary Note 4.

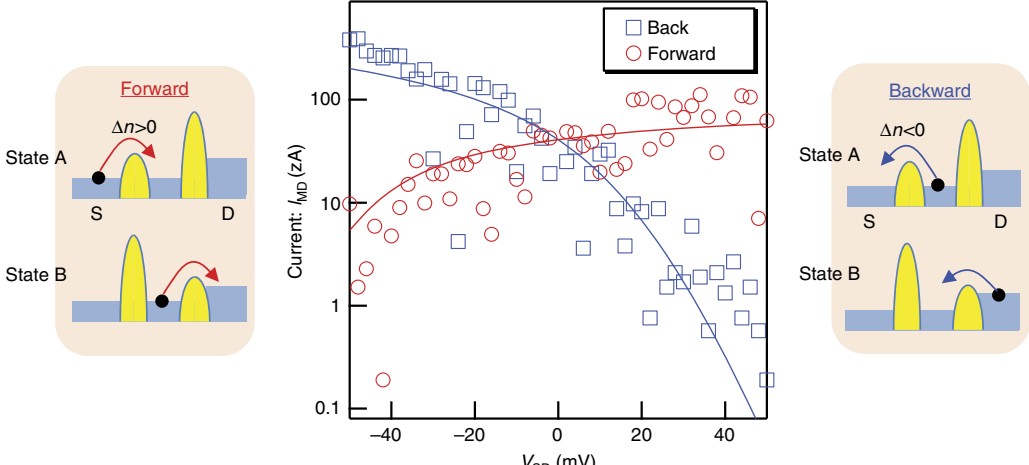

**Figure 3 | Current generation by Maxwell's demon as a function of the source–drain bias voltage $V_{SD}$ ($= V_S - V_D$).** Open dots are experimental results. The solid lines are data obtained from Monte-Carlo simulation, whose details are explained in Supplementary Note 2. The two insets show schematics for forward current (drain-to-source current obtained with $\Delta n = 2$) and backward current (source-to-drain current obtained with $\Delta n = -2$) when negative $V_{SD}$ is applied. The reason the characteristics between forward and backward currents are asymmetric originates from the difference in the terminal from which electrons are supplied: In the forward direction, electrons are supplied from the S, whose chemical potential is fixed; in the backward direction, electrons are supplied from the D, whose chemical potential is varied by $V_{SD}$. In the experiments, we applied $V_{UG} = 3.1$ V, $V_{G1} = -2.4$ V and $V_{G2} = -1.95$ V for state A; $V_{UG} = 2.7$ V, $V_{G1} = -3$ V and $V_{G2} = -1.6$ V for state B; $V_{ED} = 1$ V, $V_S = -0.6$ V for the both states.

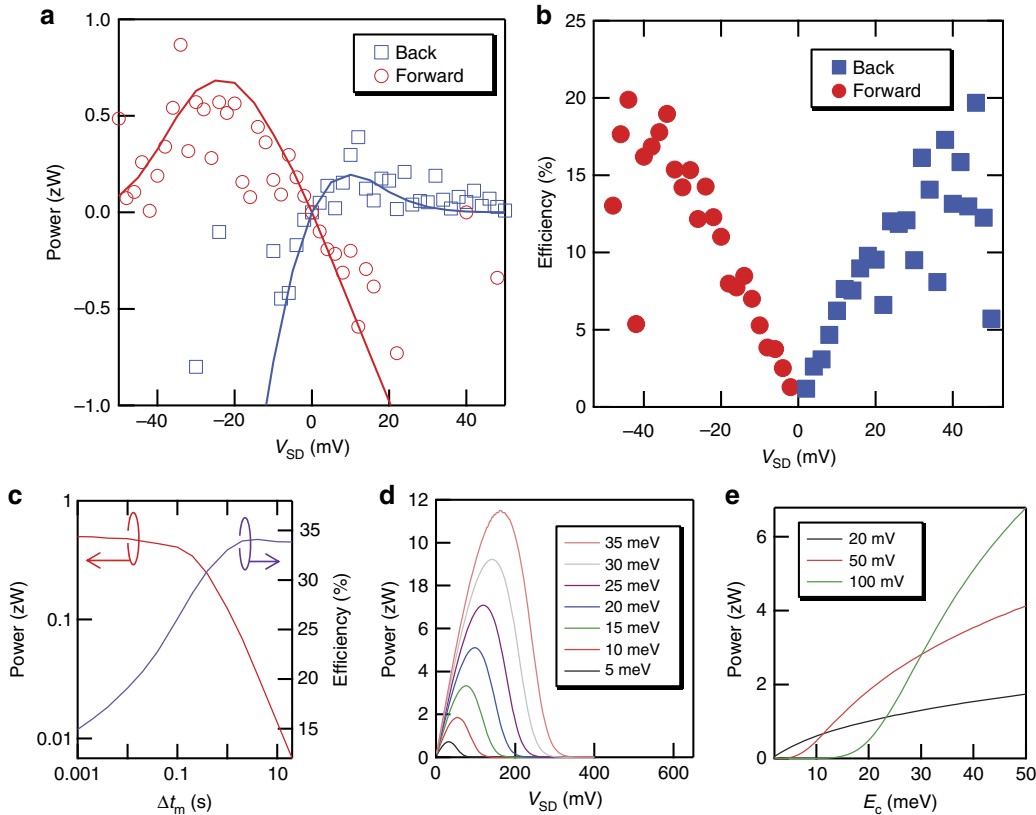

**Figure 4 | Power generation by the demon.** All voltages are the same as in the case in Fig. 3. (**a**) Generated power as a function of $V_D$. The solid lines are data obtained from Monte-Carlo simulation. (**b**) Efficiency of information-to-energy conversion as a function of $V_{SD}$. (**c**) $\Delta t_m$ dependence of the power and efficiency in Monte-Carlo simulation. (**d,e**) Numerical calculations of power generated by Maxwell's demon. Generated power as a function of (**d**) $V_{SD}$ at various $E_C$ and (**e**) $E_C$ at various $V_{SD}$. For all calculations, $k_B T = 26$ meV, $\Gamma_1 = 1\,\mathrm{s}^{-1}$, $\Gamma_2 = 10^{-10}\,\mathrm{s}^{-1}$ and $\Delta n = 1$ (Supplementary Note 5).

**Efficiency of the demon's power generation**. The efficiency is defined as the ratio of the generated energy to the mutual information, which represents conversion efficiency from information to energy. The efficiency of information-to-energy conversion is given by $\langle \Delta F \rangle / k_B T I$, where $\Delta F$ is the change in the free energy and $I$ is the mutual information, and $\langle . \rangle$ depicts the mean per one measurement. When $\Delta n_t$ electrons are transferred from the source to drain, $\Delta F$ becomes $\Delta n_t e V_D$. When the number of measurement points for this electron transfer is $N_t$, $\langle \Delta F \rangle$ is given by $\langle \Delta n_t \rangle e V_D / N_t$. When there is no error in the measurements of $n$, $I$ equals Shannon entropy $H$ given by $H = \sum_i P_i \ln P_i$, where $P_i$ is the probability for the $i$th event. In our experiments, we use information about $n$: $n$ is either increased by $\Delta n_{thresh}$ (decreased by $\Delta n_{thresh}$) or not at state A (state B). From the probabilities experimentally obtained, Shannon entropies $H_A$ and $H_B$ of the measurement outcome at state A and B, respectively, are estimated. When the number of the measurement points at state A and B are $N_A$ and $N_B$, respectively, $H$ is given by $(N_A H_A + N_B H_B)/(N_A + N_B)$. We assume $I = H$, or neglect errors in measurement of $n$. Consequently, since $I \geq H$, we can estimate the lower bound of the efficiency $\Delta F / k_B T I$ based on electron counting statistics in the experiment (Supplementary Note 6).

The efficiency increases with $|V_{SD}|$ and reaches about 18% at the $V_{SD}$ that provides the maximum power (Fig. 4b). This is because the increase in $|V_{SD}|$ increases the energy gain of an electron transferred from the source to drain. The efficiency also depends on $\Delta t_m$: When $\Delta t_m$ becomes longer, the efficiency increases and then saturates (Fig. 4c). The point of this behaviour is that it reflects how much of the information obtained by each measurement is used for the feedback: Measurements in the

absence of the feedback waste the obtained information and reduce the efficiency. When $\Delta t_m$ becomes shorter (longer) than the time taken for $n$ to increase or decrease by $\Delta n$ at state A or B, respectively, the number of measurements without the feedback increases (saturates), which leads to a decrease (saturation) in efficiency (Fig. 4c; the details are explained in Supplementary Note 5). Compared to the simulated efficiency (24%) at $\Delta t_m$ of 60 ms (Fig. 4c), the efficiency of 18% obtained in our experiment at $V_{SD} = -34$ mV is reasonably large. The further increase of $|V_{SD}|$ also increases the number of measurements in the absence of the feedback, which leads to reduced efficiency. Due to these features, the efficiency cannot reach 100% like it can in an ideal Szilard's engine utilizing all of the information. On the other hand, the power decreases with increasing $\Delta t_m$ because the time interval for carrying out the feedback increases with $\Delta t_m$, which reduces $I_{MD}$. Therefore, there is a tradeoff between the efficiency and power against $\Delta t_m$ and, in this work, we adjusted it to 60 ms to ensure the reduction in the power is small and the efficiency is relatively large. Note that the origin of the power reduction is the missed detection of fast electron transitions within $\Delta t_m$. Monte-Carlo simulation indicates that the power reduction caused by detection errors is < 10% in this work.

## Discussion
Numerical calculations indicate another way to increase the power generated by the demon (details are explained in Supplementary Note 2). For simplicity, we assume infinitely fast sensing and feedback and $\Delta t_m = 0$. In the case of Szilard's engine, the generated energy is given by $k_B T \ln 2$, which is proportional to

temperature. In our feedback with transistors, in which electrons are transported by thermal hopping, the generated power depends on another factor, which is the charging energy ($E_C = e^2/2C$) of the SEB. The power increases monotonically with $E_C$ (Fig. 4c,d). There are mainly two reasons for this behaviour of the generated power. The first is that the electron can gain energy multiples of $2E_C$ when it enters the SEB from the source, which allows the electron to reach the drain highly biased and thus to gain larger energy. The second is related to how often this energy gain happens at larger $E_C$. As explained above, the transition rate for electrons in the source to enter the SEB is independent of $E_C$. This is in contrast to an SEB sandwiched by tunneling barriers: Larger $E_C$ reduces this transition rate due to the Coulomb-blockade effect, which leads to a reduction of power in spite of larger energy gain. In other words, Maxwell's demon exploits transistors' benefits as an increase in power by increasing $E_C$, that is, decreasing $C$, which can be achieved by miniaturization technology for current transistors. We should note that large $E_C$ shortens the time that an electron stays in the SEB and tends to require faster sensing and feedback. At $\Delta t_m = 60$ ms, $E_C$ should be smaller than 35 meV to monitor the desired electron motion and perform the feedback before the electron's departure from the SEB; otherwise, $n$ changes before the feedback is completed.

The transistors provide a benefit to the generated power in this work and also have other merits. Since transistors can act as gates partitioning electrons, which have not been achieved with other systems, the procedure for the demon's power output is relatively simpler than in previous work. In addition, with an electrical approach, the transistors can not only control the chemical potential and size of the box[28,29], in which an electron is confined, but also form tunnel barriers[30] and a couple of dots[31,32]. Operation in a wide range of temperatures, including room temperature as in this work, also deepens the analysis and applications of Maxwell's demon. These features promise a experimental platform on which the demon plays active roles, such as a quantum Szilard engine[33], noise squeezer[21,34] and nano-scale heat engine[35–38]. Transistors also have another perspective supported by nanotechnology, which continues to make commercially available transistors smaller and smaller. Further shrinkage of transistors will further improve the controllability of electron motion and charge sensitivity and increase $E_C$, all of which will lead to improvement in the demon's performance. Therefore, we believe that Maxwell's demon in transistors continue to improve in performance and find new applications.

## Methods

**Device fabrication and structure.** The device is fabricated from a silicon-on-insulator (SOI) wafer. First, silicon nanowire channels for a SEB, source, drain and sensor are formed on an SOI layer with boron concentration of $10^{15}$ cm$^{-3}$, followed by thermal oxidation. The width and thickness of SOI channels for the SEB and sensor are around 30 and 20 nm, respectively. Oxide thickness is 38 nm. Then, two gates, G1 and G2, composed of poly-crystalline silicon are formed on the SOI channel between the SEB and source/drain, followed by oxidation. G2 is designed to be larger than G1 to reduce capacitive coupling between the SEB and drain and suppress changes in the chemical potential at the SEB when voltage applied to the drain is changed. However, there is a risk that small electron traps are unintentionally formed under a large gate due to the structural fluctuation of the transistor channel. Therefore, we use a small G1 to mitigate the risk and a larger G2 to reduce the potential change. The width of G1 and G2 are 30 and 200 nm, respectively. Then the whole area is covered with a 50-nm-thick oxide interlayer formed by a chemical vapour deposition. Finally, another gate (UG) is formed on the whole area. The UG is used to control the chemical potential of the SEB, induce electrons in the source and drain, and to control current flowing through the channel of the sensor.

**Data availability.** The data that support the findings of this study are available from the corresponding author on request.

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

## Acknowledgements

We appreciate comments on this manuscript from I. Mahboob of NTT Basic Research Laboratories.

## Author contributions

K.C. and K.N. conceived the experiments, analysed the data, performed the simulations and prepared the manuscript. K.C. and S.D. performed the measurements. K.N. fabricated the devices and performed the numerical calculations. All authors discussed the results and commented on the manuscript. A.F. supervised the project.

## Additional information

**Competing interests:** The authors declare no competing financial interests.

