## [Peer Review File · Nature Communications]

Reviewers' comments:

Reviewer #1 (Remarks to the Author):

SYNOPSIS:

The paper reports on the experimental implementation of a Maxwell demon based on a single electron box (SEB) that is tunnel-coupled via two transistors to source and drain electrodes. With an additional point contact, the charge state of the SEB is read out, fed into a feedback circuit which adjusts the gates separating the SEB from source and drain. Thereby, a protocol is used that rectifies transport and which can be used to produce positive electric power, effectively driving electrons upwards a potential bias using ideally only information. It is claimed that the present implementation is superior to previous approaches in that the undesired injection of work can be explicitly minimized.

--

GENERAL ASSESSMENT:

An experimental implementation of a Maxwell demon is certainly a very interesting achievement for a broad community. However, it must be said that based on single-electron counting, there have already been a number of similar proposals and implementations using very similar protocols (charge detection using quantum point contact, external feedback loop, conditioned modification of tunneling rates, effective driving of electrons against the bias), some of which are part of the reference list. Therefore, I would doubt whether the paper is innovative enough to be of interest to a broader community. The main difference of the current paper compared to previous single-electron implementations is claimed to be that the used feedback operations in principle do not inject any work into the system and is thus closer to a true Maxwell demon. In my opinion, this claim is not sufficiently proven. In addition, the paper remains unclear also in several technical points. Therefore, I would not recommend publication of the paper.

CRITICISM:

1.)

The authors seem to use/assume that the SEB is in some thermal state with an adapted chemical potential, that depends on the number of excess electrons in it. Why should it be in a thermal state at all and not in some non-equilibrium steady state?

If there is some thermalizing process (e.g. heat exchange with background phonons), this heat will affect the energy balance and should be taken into account.

It is hard to follow the claims that the feedback does not inject work if the energies of the system are not known

microscopically but rather estimated from a classical picture.

2.)

How reliable is the charge detection scheme? Fig. 1c shows a quite jumpy trajectory, so it may be that with the stated measurement interval of 60ms the device may miss some electrons entering and leaving the SEB. The authors should provide some estimate for such errors.

3.)

Unfortunately, all experiments have been performed at room temperature only. It would be nice to demonstrate the dependence on temperature to make the action of a Maxwell demon more explicit.

Reviewer #2 (Remarks to the Author):

The paper reports on a nice experimental realization of a Maxwell demon using a single electron box. In my opinion, the experiment does not provide any new significant information about the thermodynamics of information. However, I still think that it is a relevant experiment because it goes beyond the previous realizations of Maxwell demon in some qualitative aspects. The main one is that the energy extracted as work is given by a true electric power, i.e., a charge current flowing against some voltage. In previous experiment the work was extracted by the manipulation of some external parameter, like decreasing the intensity of an optical trap, which is similar to the expansion of a gas against a piston. We know how to use the pressure exerted on the piston to move gears, but it is not clear how to use the energy released to an optical trap when the intensity is decreased. In the present experiment, the extracted work is a normal electric power.

I think this novelty is interesting enough to draw the attention of many researchers and inspire new experiments on the subject.

However, the manuscript has a serious problem, which is the poor presentation.

The introduction is not clear enough for the average reader and some sentences do not seem grammatically correct, like "can grow the demon's performance" (lines 18-19) or obscure, like "exert work onto the objects" (lines 26-27).

The question "What is the energetical origin of the generated energy?" (line 28) is misleading. The energy comes from the thermal bath. There is no debate on this issue.

The forward and the backward operation is not described in the text. The only description is in figure 3, which is not clear. I guess that the backward operation corresponds to negative Δn (line 62), but it is not said explicitly in the text. In fact, it is not clear the meaning of negative Δn , nor the relationship between Δn and k .

The sentence in lines 109-110, "since more power is generated and efficiency is higher than when..." makes no sense. The role of Δn is not clear at all along the text.

There should be more information in the text about the informational aspects of the experiment: measurement outcomes, mutual information, etc.

The scale of vertical axis in Fig. 3 is in contradiction with the main text (line 111). Units are inconsistent in figures 4c

The equations in the supplementary material are had to follow, specially (7) and (8). The authors should use a better equation editor.

In summary, the experiment is nice but the manuscript is really hard to read in its present form.

Reviewer #3 (Remarks to the Author):

SUMMARY OF THE KEY RESULTS

A physical realisation of a Maxwell's demon is presented. It is based on rectifying the thermal motion of electrons so that they flow in one direction, creating a net current. This is done by having a bottle neck electron 'island' with gates on either side (field effect transistors which are opened and closed by application of an external electric field). The gates are opened or closed depending on the result of monitoring the occupation number of electrons on the island in real time. When the number goes up the gate to the source is closed and the one to the drain is opened; thus there is a net flow to the drain.

ORIGINALITY AND INTEREST

To my knowledge the specific design is new even though it shares some features with others. Using electron islands and monitoring occupancy in real time to realise a kind of Szilard engine was done in their reference 19 by the Aalto group. A key difference is that in this paper under review the power is actually extracted. The idea of a ratchet demon is of course also not new. I personally find this type of demon very interesting and have wondered whether it could be made so was particularly interested in seeing this paper. It provides a nice link between information thermodynamics and modern technology which is of course heavily based on silicon electronics.

DATA & METHODOLOGY: VALIDITY OF APPROACH, QUALITY OF DATA & PRESENTATION

The presentation is overall very understandable without unnecessary technicalities. Certain introductory explanations could be better as described below. I do not see reasons to question the quality of the data.

APPROPRIATE USE OF STATISTICS AND TREATMENT OF UNCERTAINTIES

This paper has statistics at its heart and deals with them very explicitly.

CONCLUSIONS: ROBUSTNESS, VALIDITY AND RELIABILITY

The experimental data is plausible given their explanations as well as the existence of related experiments mentioned above, which indicates that it is indeed possible to perform the experiment in question. The interpretation as a Maxwell's demon is sound and made clear through the link the authors make explicitly between mutual information and the work extracted.

SUGGESTED IMPROVEMENTS

In the introduction, when describing why the demon does not violate the second law, it would be pedagogical to describe Kelvin's second law which refers to closed cycles and is the relevant one here.

REFERENCES

The references are broadly done acceptably. The referencing suggests that the relatively recent

information thermodynamics references resolved the demon paradox but this should be clearly attributed to Bennett. It should also be acknowledged in the intro that some other papers have actually extracted work with the demon, including -I think- their references 14 and 19.

CLARITY AND CONTEXT

As mentioned above the presentation is clear. The title, abstract, introduction (modulo the comments above) and conclusions are appropriate.

With the above in mind, I recommend the manuscript for publication in Nature Communications, subject to the requested minor changes in the referencing.

Reply to Reviewer #1

Thank you very much for your critical comments. As you mentioned, there are nice experiments that used metal-based single electron devices to convert information into energy with single-electron detection; however, our work is different from previous reports, in that we used silicon transistors to take advantage of their ability to tune electron transition rates with gate voltages over decades, which is difficult with metal-based devices. Thanks to this characteristic, we were able to demonstrate the demon's door operation and output electric power. With this work, we want to show that silicon transistors are ideal platforms for studying information thermodynamics, such as a quantum Szilard engine (31) and nano-scale heat engines (32–35). We believe that this work, performed with silicon transistors, will make information thermodynamics more accessible to researchers in many fields because silicon transistors are such common devices in the world today.

In the following, we respond to your comments.

[Your comment (1)]

1.)The authors seem to use/assume that the SEB is in some thermal state with an adapted chemical potential, that depends on the number of excess electrons in it. Why should it be in a thermal state at all and not in some non - equilibrium steady state? If there is some thermalizing process (e.g. heat exchange with background phonons), this heat will affect the energy balance and should be taken into account.

[Answer]

Thank you very much for pointing this important phenomenon out. Before discussing non-equilibrium phenomena, let me explain why we consider our SEB to be at equilibrium. In our previous work [K. Nishiguchi et al., Nanotechnology 25, 275201 (2014)], we observed thermal fluctuation in the number of electrons in the SEB (n) and obtained the relation $\sigma^2 * E_C = k_B T / 2$, where σ^2 is variance of n , $E_C (= e^2 / 2C)$ is the charging energy of the SEB, k_B is Boltzmann's constant, and T is temperature. We obtained the same relation at 300 K with the devices used in this work. In addition, we obtained the same thermal fluctuation in the distribution of the number of transferred electrons by one cycle of periodic control. This means that the SEB is at equilibrium even under the feedback control (not shown). If there were thermalizing effects, the distribution would differ from the thermal fluctuation. Note that our observation is similar to that in Fig.2(a) in a previous work [B. Kung et al., Phys. Rev. X 2, 011001 (2012)].

In our experiment, as mentioned above, thermalizing effects were not observed. In addition, the Monte-Carlo simulations we performed without considering thermalizing effects reproduce our experimental results very well. The reason is as follows: since our experiment is performed at room temperature, non-equilibrium effects with an energy scale smaller than 26 meV are not visible. Therefore, we ignored them in this work.

[Your comment (2)]

It is hard to follow the claims that the feedback does not inject work if the energies of the system are not known microscopically but rather estimated from a classical picture.

[Answer]

Thank you very much for this comment. In response, we need to explain the mechanism of single-electron ratchet from its stochastic aspect that originates from the microscopic picture.

If work is exerted onto the electrons in the SEB by feedback, the work turns into a shift in the chemical potential in the SEB, $\Delta\mu_{\text{SEB}}$. A change in voltage applied to transistors causes $\Delta\mu_{\text{SEB}}$ because of capacitive coupling between the transistors and the SEB (see Fig. S4a to b). If $\Delta\mu_{\text{SEB}} > 0$, the SEB become non-equilibrium and the excess electrons in it exit during equilibration (see Fig. S4b to c). The number of transferred electrons, Δn_t , fluctuates among trials because of thermal fluctuation (see Fig. A). This stochastic Δn_t is a characteristic of the microscopic aspect of the mechanism. To determined $\Delta\mu_{\text{SEB}}$, we calculated I_{SEB} as the average of Δn_t in the period of one second.

If work is not exerted onto electrons in the SEB, Δn_t fluctuates around zero and the average, I_{SEB} , is zero. Note that the direction of I_{SEB} is determined by the sign of $\Delta\mu_{\text{SEB}}$ (see Fig. A).

Fig. A Schematics of stochastic (microscopic) nature of single-electron ratchet mechanism.

Control of transistors from state A to B changes μ_{SEB} because of the capacitive coupling between them and the SEB. As a result, after the control, Δn_t electrons transit from the SEB to the drain. Since electrons are thermally fluctuating, Δn_t fluctuates among trials. Probability p of Δn_t distributes as shown in (a) when $\Delta\mu_{SEB} > 0$, (b) when $\Delta\mu_{SEB} = 0$, and (c) when $\Delta\mu_{SEB} < 0$. As for control of transistors from state B to A, Δn_t electrons transit from the source to the SEB (see Fig. 4S c to d). The ratchet current I_{SEB} is obtained by repeating the controls. Under periodic control, I_{SEB} can be represented by $e \times \langle \Delta n_t \rangle / \Delta t_{cyc}$, where e is the elementary charge, $\langle \Delta n_t \rangle$ is the ensemble average of Δn_t over the trials, and Δt_{cyc} is time consumed for one cycle of the controls. Since the addition of an electron into the SEB increases μ_{SEB} by e^2/C , $\Delta\mu_{SEB} = \langle \Delta n_t \rangle \times 2E_C$, we can derive $\Delta\mu_{SEB}$ from I_{SEB} , where $E_C (= e^2/2C)$ is the charging energy of the SEB.

[Your comment (3)]

2.) *How reliable is the charge detection scheme? Fig. 1c shows a quite jumpy trajectory, so it may be that with the stated measurement interval of 60ms the device may miss some electrons entering and leaving the SEB. The authors should provide some estimate for such errors.*

[Answer]

Thank you very much for the important question. We can estimate the effect of detection errors from Fig. 4c. By decreasing measurement interval Δt_m , maximum power P_{MAX} saturates at 0.42 zW, this value corresponds to power output without the errors. With $\Delta t_m = 60$ ms, we obtain $P_{MAX} = 0.38$ zW. This means that the effect of the errors is 10% at most at the condition for maximum power output in our experiment. As you expected, the error increases and P_{MAX} decreases when Δt_m increases. Note that we choose $\Delta t_m = 60$ ms to obtain relatively large P_{MAX} and efficiency under the tradeoff between them (see Fig. 4c).

[Your comment (4)]

3.) *Unfortunately, all experiments have been performed at room temperature only. It would be nice to demonstrate the dependence on temperature to make the action of a Maxwell demon more explicit.*

[Answer]

Thank you very much for the nice suggestion. With our system, we can perform temperature-dependence measurement. However, with this manuscript, we want to show that silicon transistors are ideal platforms for studying information thermodynamics by using their advantage of transition rate control with gate voltage. For this purpose, we believe that room-temperature measurement is the best choice. In addition, we have studied the thermal fluctuation in n at room temperature, and our experimental results are well reproduced by Monte-Carlo simulation. They strengthen our claim that we achieved power generation with a Maxwell demon that derives energy from a thermal bath.

Naively, temperature-dependence measurement would just show the proportional increase in the output energy with temperature since the demon derives energy from a thermal bath. This can be assumed from our previous study [K. Nishiguchi et al., *Nanotechnology* 25, 275201 (2014)], which shows a proportional increase in the average energy of electrons in the SEB with temperature. However, the temperature dependence of the power output is complicated because the transition rate of electron hopping and n depends on temperature. Therefore, we do not expect that we can relate power output to temperature with a simple explanation. We tried to estimate the temperature dependence of power output by Monte-Carlo simulation, but we still do not have clear explanation

of the results. In addition, low-temperature measurement probably produces back actions and other non-equilibrium phenomenon with a small energy scale.

We thank you very much for your interesting suggestion. However, since a temperature-dependence experiment would probably provide us with complicated information that is beyond our claim in this manuscript, temperature-dependence measurement remains for the future as a separate work.

We hope our responses are appropriate.

Sincerely yours,
Kensaku Chida

Reply to Reviewer #2

First of all, we apologize for bad writing and lack of information in our manuscript. To improve this point, we asked colleagues who are native English speakers to comment on this manuscript, and we reflect their comments in it.

We thank you very much for evaluating our work as “interesting enough to draw the attention of many researches and inspire new experiments in the subject”. Your comments strongly encourage us. In addition, thanks to your kind comments, we were able to greatly improve the readability of our manuscript and construct more scientifically appropriate expressions.

In the following, we respond to your comments.

[Your comment (1)]

The introduction is not clear enough for the average reader and some sentences do not seem grammatically correct, like "can grow the demon's performance" (lines 18 - 19) or obscure, like "exert work onto the objects" (lines 26 - 27).

[Answer]

Thank you very much for your kind reading. We changed those sentences and the sentences have been checked by native English speaking colleagues.

[Your comment (2)]

The question "What is the energetical origin of the generated energy?" (line 28) is misleading. The energy comes from the thermal bath. There is no debate on this issue.

[Answer]

Thank you very much for your professional comment. We deleted this misleading question.

[Your comment (3)]

The forward and the backward operation is not described in the text. The only description is in figure 3, which is not clear. I guess that the backward operation corresponds to negative Δn (line 62), but it is not said explicitly in the text. In fact, it is not clear the meaning of negative Δn , nor the relationship between Δn and k .

[Answer]

Thank you very much for pointing this lack of information out. We have clearly stated the relationship between Δn and current direction in the text and have added sentences to explain the difference between Δn and k .

[Your comment (4)]

The sentence in lines 109 - 110, "since more power is generated and efficiency is higher than when..." makes no sense. The role of Δn is not clear at all along the text.

[Answer]

Thank you very much for pointing this lack of information out. We assessed the relationship between Δn and the output power by Monte-Carlo simulation. Δn was determined by the threshold value of I_{det} (ΔI_{det}) in the experiment. Since the measurement becomes unstable with small ΔI_{det} and output power becomes small with large ΔI_{det} , we performed all of the measurement with $\Delta n(\Delta I_{\text{det}}) = 2$. As we did not show the role of Δn in our experiments, we just mention our experimental conditions in the revised manuscript. We try to experimentally clarify the role of Δn in the future.

[Your comment (5)]

There should be more information in the text about the informational aspects of the experiment: measurement outcomes, mutual information, etc.

[Answer]

Thank you very much for your comment. We added an explanation about the informational aspects of the experiment in the text, which was described in the supplementary materials section 5 in the previous manuscript.

[Your comment (6)]

The scale of vertical axis in Fig. 3 is in contradiction with the main text (line 111). Units are inconsistent in figures 4c.

[Answer]

We are sorry for those inconsistencies and appreciate your pointing them out. We fixed them.

[Your comment (7)]

The equations in the supplementary material are had to follow, specially (7) and (8). The authors should use a better equation editor.

[Answer]

Thank you very much for suggesting we use a better equation editor. We wrote all supplementary materials with TeX, which greatly improves readability of the supplementary materials.

Your comments helped us to improve our manuscript significantly. We hope our responses and revisions are appropriate.

Sincerely yours,

Kensaku Chida

Reply to Reviewer #3

We are grateful for your recommending our manuscript for publication in *Nature Communications*. Your comments give us confidence in our work and helped us make the manuscript appropriate for a broad readership.

[Your comment (1)]

In the introduction, when describing why the demon does not violate the second law, it would be pedagogical to describe Kelvin's second law which refers to closed cycles and is the relevant one here.

[Answer]

Thank you very much for your kind suggestion. We added sentences mentioning Kelvin's second law in the introduction.

[Your comment (2)]

The references are broadly done acceptably. The referencing suggests that the relatively recent information thermodynamics references resolved the demon paradox but this should be clearly attributed to Bennett. It should also be acknowledged in the intro that some other papers have actually extracted work with the demon, including - I think - their references 14 and 19.

[Answer]

Thank you very much for pointing out these details that we overlooked. We added Bennett's work in the references and a sentence to mention previous works that have extracted work with the demon.

We hope our responses are appropriate.

Sincerely yours,
Kensaku Chida

List of the revisions

1. According to comment (3) from reviewer #1, we added a sentence mentioning the detection errors in the last part of the paragraph starting with “The efficiency increases with $|V_{SD}|$...”.
2. According to comment (1) from reviewer #2, we changed words in the last sentence in the abstract and forth sentence in the paragraph starting with “One famous example of...”.
3. According to comment (2) from reviewer #2, we deleted the last sentence in the paragraph starting with “One famous example of...”.
4. According to comment (3) from reviewer #2, we added parenthesis in the 5th and 6th sentence in the paragraph starting with “When G1 opens and G2 closes...”, 1st and 2nd sentences in a paragraph starting with “With the feedback,”, and 4th sentence in the legend for Fig. 3.
5. According to comment (4) from reviewer #2, we modified from the third to fifth sentences of the paragraph starting with “Power generation by Maxwell’s demon”.
6. According to comment (5) from reviewer #2, we moved the discussion of the information aspect of the experiment from supplementary material section 5 to the paragraph starting with “Efficiency of the demon’s power generation”.
7. According to comment (6) from reviewer #2, we fixed the units in Fig. 3 and Fig. 4e.
8. According to comment (7) from reviewer #2, we wrote the supplementary materials with TeX.
9. According to comment (1) from reviewer #3, we added a sentence mentioning the Kelvin’s second law in the last part of the paragraphs starting with “One famous example of...” and “This question has been answered...”.
10. According to comment (2) from reviewer #3, we added one reference.
11. According to comment (2) from reviewer #3, we added a sentence mentioning previous works that has extracted work after the third sentence of the paragraph starting with “In addition to this theoretical progress...”.

Reviewers' comments:

Reviewer #1 (Remarks to the Author):

SYNOPSIS:

The authors have done some improvements to their article. However, I still doubt whether the paper is innovative enough

to be of interest for a large community. In my opinion, the authors still miss a microscopically correct model of the device

and therefore cannot fully claim that they can quantify the entropic balances correctly, particularly important for a

discussion of a Maxwell demon.

While the work certainly warrants publication somewhere after suitable clarifying discussions (further comments below),

I do not find it suitable for Nature Communications.

CRITICISM:

1.)

For non-autonomous systems with an external feedback loop, additional sources of entropy production need to be considered.

One is as previously mentioned the energy injected during control operations: I am concerned that just observing a vanishing mean

current for open-loop control when $\mu_S = \mu_D$ might not be interpreted as proof of vanishing energy injection during the feedback

as the contributions from the two gates may just cancel, this would when source and drain potentials are different lead to an

imbalance between energy currents entering the SEB and affect the total entropy production.

Another one is the required entropy production for the measurement. As e.g. discussed in PRL 114, 158101 (2015), a huge amount of

heat is dissipated already to perform the measurement. Furthermore, the measurement device may itself inject energy into the system.

Finally, there is also a huge amount of entropy created in the classical processing of the result.

These effects cannot be completely excluded but need to be properly discussed, in particular when a wide audience is to be addressed.

2.)

For finite-size system such as a SEB, there are differences between the canonical ensemble and the grand-canonical treatment with

a time-dependent chemical potential approximately reflecting the excess particle number on the SEB.

While these differences can be expected to vanish in the thermodynamic limit, for finite-sized systems they are still important.

Without proper discussion, it is unclear whether the heat exchanges with source and drain are quantified appropriately.

3.)

I still find the article quite unclear in many other definitions. For example, when discussing the information-theoretic efficiency of the

device, the authors compare the free energy with the mutual information between detector and

system. The latter will fluctuate in time as well, the authors estimate 10% error probability but calculate with perfect detection, so it is not clear what the authors mean by the mutual information, its average value? A rigorous discussion of these effects would require to clearly fix the states of the system and of the detector and to consider a model for erroneous measurements.

Reviewer #2 (Remarks to the Author):
See attached

Reviewer #3 (Remarks to the Author):

I am still in favour of publishing the manuscript, for the reasons given in the previous report.

This below paragraph concerning what is novel about the result still needs modification:

"Some previous experiments have succeeded in extracting work with the demon 15, 20. However, since the generated free energy in these demonstrations is confined to small systems, converted to potential energy or a temperature difference, or is inaccessible from the outside, the demon's next task is to make the generated energy accessible, i.e., to output electric power."

This paper (one of their references I mentioned) appears to generate electric power.

Vidrighin, M. D. et al.
Photonic Maxwell's demon.
Phys. Rev. Lett. 116, 050401 (2016).

My impression is rather that the novelty is that this is one of the most natural possible demons that has now been realised: a demon that rectifies electric current to output electrical power.

Reply to Reviewer #1

Thank you very much for stating that our manuscript certainly warrants publication somewhere after suitable clarifying discussions. Thanks to your criticisms, our discussion has become deeper and more appropriate.

In the following, we respond to your comments.

[Your comment (1)]

1.) For non - autonomous systems with an external feedback loop, additional sources of entropy production need to be considered. One is as previously mentioned the energy injected during control operations: I am concerned that just observing a vanishing mean current for open - loop control when $\mu_S = \mu_D$ might not be interpreted as proof of vanishing energy injection during the feedback as the contributions from the two gates may just cancel, this would when source and drain potentials are different lead to an imbalance between energy currents entering the SEB and affect the total entropy production.

[Answer]

Thank you very much for your comment. When source and drain potentials are different, the imbalance in energy current leads to additional entropy production, which is due to undesirable electron motion over the higher potential barrier, for instance, electron motion between the SEB and drain at state A. In our experiment, ~2% of electron motion is undesirable. This undesirable electron motion reduces ~2% of the demon's power generation. We discuss the undesirable electron motion in Supplementary Section 3.

[Your comment (2)]

Another one is the required entropy production for the measurement. As e.g. discussed in PRL 114, 158101 (2015), a huge amount of heat is dissipated already to perform the measurement. Furthermore, the measurement device may itself inject energy into the system. Finally, there is also a huge amount of entropy created in the classical processing of the result. These effects cannot be completely excluded but need to be properly discussed, in particular when a wide audience is to be addressed.

[Answer]

Thank you very much for this interesting information. We added a paragraph in Supplementary Section 1 based on your comment. Your comment strengthens our discussion.

[Your comment (3)]

2.) *For finite - size system such as a SEB, there are differences between the canonical ensemble and the grand - canonical treatment with a time - dependent chemical potential approximately reflecting the excess particle number on the SEB. While these differences can be expected to vanish in the thermodynamic limit, for finite - sized systems they are still important. Without proper discussion, it is unclear whether the heat exchanges with source and drain are quantified appropriately.*

[Answer]

Thank you very much for the comment. We measured time traces of the number of electrons in the SEB, and treated the snapshots as ensembles. Our experimental results—the ensembles with mean electron number—are consistent with the canonical ensembles. A previous study [A. Hofmann et al., Phys. Rev. B 93, 035425 (2016).] also treated the ensembles in a similar way. To answer completely to your comment, we need further theoretical consideration, which will be future work with theoretical supports.

We consider that we appropriately evaluate the heat exchange accompanying the demon's power generation because we have traces of the number of electrons in the SEB, tunable barriers formed with transistors, and Monte-Carlo simulations consistent with experimental results.

[Your comment (4)]

3.) *I still find the article quite unclear in many other definitions. For example, when discussing the information - theoretic efficiency of the device, the authors compare the free energy with the mutual information between detector and system. The latter will fluctuate in time as well, the authors estimate 10% error probability but calculate with perfect detection, so it is not clear what the authors mean by the mutual information, its average value? A rigorous discussion of these effects would require to clearly fix the states of the system and of the detector and to consider a model for erroneous measurements.*

[Answer]

Thank you very much for your comment. Since we want to demonstrate a proof-of-concept experiment on Maxwell's demon, we tried to exclude effects that give us advantages. We neglected measurement errors so that we could maximize the mutual information and get a lower bound of the efficiency of information-to-energy conversion.

As you recommended, we consider a model for erroneous measurements [J. Koski et al., Phys. Rev. Lett. 113, 030601 (2014).]. We have added a supplementary section to discuss it.

We found a theoretical paper that discusses the balance of entropy production, the canonical ensemble and the grand-canonical ensemble, and measurement errors in setup similar to the one in our work [G. Schaller, ArXiv: 1612.04167]. This theoretical work probably addresses your concerns.

Thanks to your comments, our discussion is now through enough for our paper to be published. We believe our work is now suitable for *Nature Communications*. We hope our responses are appropriate.

Sincerely yours,
Kensaku Chida

Reply to Reviewer #2

We thank you very much for recommending our manuscript for publication in Nature Communications. We are very happy to hear that you are satisfied with our revision. In addition, your suggestions improve the readability of the manuscript for a wide readership. We are grateful to you for suggesting those changes for publication.

In the following, we respond to your suggestions.

[Your comment (1)]

The abstract still need some rewriting in my opinion. There is a typo "fluctuations" in line 9. Two sentences are not clear to me: 1) "pointed to the reality of Maxwell's demon" (line 19), and 2) "gates partition an electron's trajectory alternately". These sentences should be rephrased in a clearer way.

[Answer]

Thank you very much for your kind reading. We changed those sentences, and the sentences have been checked by native English speaking colleagues.

[Your comment (2)]

The authors use Δn for two different quantities: the deviation of the number of electron from its initial value and the threshold above which the gate voltages change. This is very confusing. There should be two different symbols for these two quantities, like Δn_t and Δn_{thresh} .

[Answer]

Thank you very much for your kind suggestion. We changed Δn to Δn_t and Δn_{thresh} .

[Your comment (3)]

Figure 1. The caption mentions an upper gate UG and a voltage VUG which cannot be seen in the figure. This gate and the corresponding voltage are explained in the text. However, if they do not appear in the figure, any reference to UG and VUG in the caption should be removed. Another (preferable) option is to include the gate and its voltage in the scheme (fig. b).

[Answer]

Thank you very much for suggesting this option. We include the upper gate and its voltage in Fig. 1b.

Your comments helped us to improve our manuscript significantly. We hope our responses and revisions are appropriate.

Sincerely yours,
Kensaku Chida

Reply to Reviewer #3

We are grateful for your recommending our manuscript for publication in *Nature Communications*, again. Your comment made our manuscript more scientifically appropriate.

In the following, we respond to your comment.

[Your comment (1)]

This below paragraph concerning what is novel about the result still needs modification: "Some previous experiments have succeeded in extracting work with the demon 15, 20. However, since the generated free energy in these demonstrations is confined to small systems, converted to potential energy or a temperature difference, or is inaccessible from the outside, the demon's next task is to make the generated energy accessible, i.e., to output electric power."

This paper (one of their references I mentioned) appears to generate electric power.

Vidrighin, M. D. et al.

Photonic Maxwell's demon.

Phys. Rev. Lett. 116, 050401 (2016).

My impression is rather that the novelty is that this is one of the most natural possible demons that has now been realised: a demon that rectifies electric current to output electrical power.

[Answer]

Thank you very much for pointing this out. We modified the sentences based on your kind comments.

We hope our responses are appropriate.

Sincerely yours,

Kensaku Chida

List of the revisions

1. According to comment (2) from reviewer #1, we added a reference and a paragraph mentioning the parasitic energy consumption in Supplementary Section 1.
2. According to comment (4) from reviewer #1, we added a supplementary section to discuss the detection errors.
3. According to comment (1) from reviewer #2, we modified expressions in the abstracts. We changed “pointed to the reality of Maxwell’s demon” to “explained the physical validity of Maxwell’s demon” and changed “gates partition an electron’s trajectory alternately” to “gates that control an electron’s trajectory”.
4. According to comment (2) from reviewer #2, we changed Δn to Δn_t and Δn_{thresh} .
5. According to comment (3) from reviewer #2, we modified Fig. 1b.
6. According to comment (1) from reviewer #3, we modified sentences in the paragraph starting with “In addition to this theoretical progress,”.
7. We added two relevant works, which were published recently, to the references.